# It's All about the Zone: Spider Assemblages in Different Ecological Zones of Levantine Caves

Jordan P. Cuff [1], Shlomi Aharon [2,3], Igor Armiach Steinpress [2,3], Merav Seifan [4], Yael Lubin [4] and Efrat Gavish-Regev [2,*]

1   School of Natural and Environmental Sciences, Newcastle University, Newcastle NE1 7RU, UK; jordancuff@gmail.com
2   The National Natural History Collections, Edmond J. Safra Campus, Giv'at Ram, The Hebrew University of Jerusalem, Jerusalem 9190401, Israel; shlomi.aharon1@mail.huji.ac.il (S.A.); igor.armiach@mail.huji.ac.il (I.A.S.)
3   Department of Ecology, Evolution and Behavior, Edmond J. Safra Campus, Giv'at Ram, The Hebrew University of Jerusalem, Jerusalem 9190401, Israel
4   Mitrani Department of Desert Ecology, Swiss Institute for Dryland Environmental and Energy Research, Blaustein Institutes for Desert Research, Sede Boqer Campus, Ben-Gurion University of the Negev, Midreshet Ben-Gurion 8499000, Israel; seifan@bgu.ac.il (M.S.); lubin@bgu.ac.il (Y.L.)
*   Correspondence: efrat.gavish-regev@mail.huji.ac.il

**Abstract:** Caves possess a continuum of ecological zones that differ in their microhabitat conditions, resulting in a gradient of nutrients, climate, and illumination. These conditions engender relatively rapid speciation and diverse assemblages of highly specialised spider fauna. It is unclear, however, how zonation of these caves affects spider assemblage composition and structure. Surveys of 35 Levantine caves were conducted to compare the assemblages of spiders between their different ecological zones. The diverse spider assemblages of these caves differed between the entrance, twilight, and dark zones, with troglophiles and accidental species occupying the cave entrance, endemic troglobites occupying the dark zones, and hybrid assemblages existing in the twilight zones. The progression of assemblage composition and divergence throughout cave zones is suggestive of processes of ecological specialisation, speciation, and adaptation of cave-endemic troglobites in the deepest zones of caves, while cave entrance assemblages are composed of relatively common species that can also be found in epigean habitats. Moreover, the cave entrance zone assemblages in our study were similar in the different caves, while the cave dark zone assemblages were relatively distinct between caves. Cave entrance assemblages are a subset of the regional species pool filtered by the cave conditions, while dark zone assemblages are likely a result of adaptations leading to local speciation events.

**Keywords:** Araneae; dark; diversity; cave entrance; hypogean; Mediterranean; southern Levant; subterranean; twilight; troglobite; troglophile

## 1. Introduction

Subterranean habitats such as caves are home to species with adaptations and pre-adaptations to darkness and nutrient limitation. Given their unique environments and reduced connectivity, caves resemble islands for obligate troglobite (obligated to life in caves) inhabitants, with little gene flow occurring between populations [1]. These conditions lead to a formation of unique assemblages of highly specialised invertebrates. The cave structure—particularly the characteristics of cave openings—has a direct effect on the abiotic conditions within the cave, such as light intensity, climatic and air conditions, and energy and nutrient influx. These abiotic conditions establish several ecological zones—entrance, twilight, and dark [2–5]—that are primarily characterised by decreasing illumination with progression into the cave.

The assemblages present in the different ecological zones of caves, however, are influenced by a myriad of factors, from specialisation, through competition and nutrient limitation, to specific microhabitat requirements [6,7]. Although caves offer relatively stable microclimates overall, cave fauna are sensitive to the changes elicited by the range of microhabitat features present throughout caves, leading to the association of some species with particular zones of caves [6,8]. Connectivity of epigean and hypogean habitats also affects assemblage composition—particularly inside the cave entrance—via colonisation and dispersal within and between the epigean and hypogean systems [9]. While it is suggested that species richness of troglophiles (species that have source populations in both hypogean and epigean habitats [10,11]) and accidentals (occasional visitors in caves) is explained by local ecological factors and seasonality, species richness of troglobites may be better explained by historical biogeography [8,11].

The Levant—a distinct biogeographical province that evolved during the Miocene—comprises the northeastern African and northwestern Arabian plates, and the eastern Mediterranean Levantine basin [12]. The Levant is positioned at a junction of three continents—Europe, Asia, and Africa—and has served as a land bridge for many terrestrial animals of different origins since its inception [13,14]. This land bridge enables the existence of four zoogeographical elements in the southern Levant *sensu stricto* (Israel, Jordan, Palestine, and the Sinai Peninsula)—Palaearctic, Ethiopian, Palaeoeremic, and Oriental [15,16]—in relation to three climate zones: Mediterranean, steppe, and desert. This biogeographic and climatic heterogeneity has resulted in a diverse regional epigean arachnid species pool [17].

The arachnofauna of caves in the southern Levant, however, is poorly characterised compared to its epigean habitats, and to that of European caves [17,18]. We have previously demonstrated that southern Levantine caves harbour diverse assemblages of troglophile and troglobite arachnids, with 62 observed spider species in 35 caves in Israel and Palestine, including 32 troglobite and troglophile spiders [19]. Previous analyses have not fully investigated differences in spider assemblage composition between different cave ecological zones, despite several species seeming to exhibit specialisations toward either the cave entrance or cave dark zones (Gavish-Regev, personal observations). In the present study, we compare the spider assemblages of different cave ecological zones of southern Levantine caves (Israel and Palestine) using various community ecological analytical approaches, including relatively recent methods that are well-suited to comparing assemblages containing the rare species typical of cave systems [20]. The application of such methods facilitates an accurate and sensitive analysis of the complex assemblages present in different cave zones. In addition, we compare the assemblages based on temperature, elevation, and geographical region. We test the following hypotheses: (1) cave zones host diverse assemblages of spiders containing many endemic species; (2) spider assemblages differ between cave zones; (3) spider assemblages exhibit a high degree of nestedness, with dark zone assemblage composition depending on the species pools of the twilight and entrance zones; (4) spider species co-occur in the same zones regularly; and (5) abiotic variables such as temperature and elevation affect cave zone spider assemblage composition.

## 2. Materials and Methods

### 2.1. Cave Surveys

Arachnids were collected from 35 caves located in Israel and Palestine (West Bank) (Figure 1) from three cave ecological zones: the cave entrance (inside the cave near its entrance, with high incidence of light); the twilight zone (in the intermediate part of the cave, when present, with low incidence of light); and the inner dark zone (beyond the twilight zone, when present, with no light), as well as outside each cave (but near to its entrance; Figure 2). The caves are distributed along the climatic gradient from the mesic Mediterranean climate in the north and centre of Israel and Palestine (12 caves in each region), to the arid and hyper-arid climates in the south of Israel (11 caves).

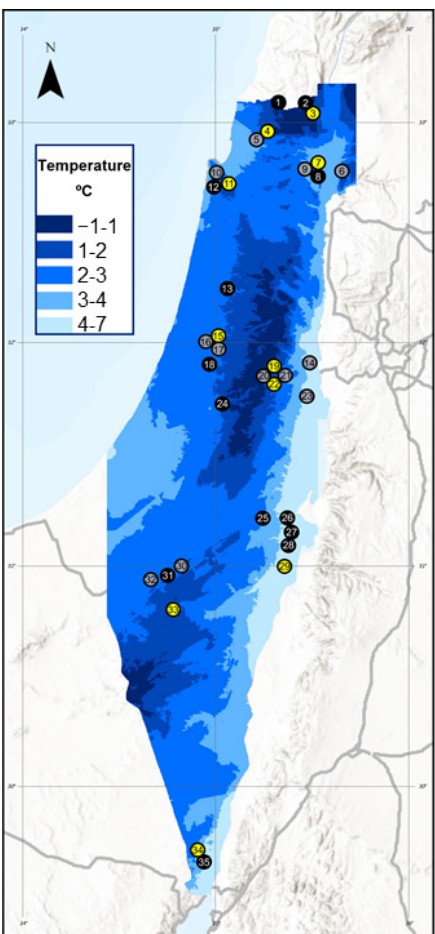

**Figure 1.** Distribution of the 35 caves included in this study. A blue overlay is given to represent the mean minimum temperature in January (the coldest month of the year) over the 20 years preceding 2021. Dark blue and light blue denote low and high minimum temperatures, respectively. Caves with both twilight and dark zones are denoted by black dots, caves with twilight zones but not dark zones are denoted by grey dots, and caves lacking both twilight and dark zones are denoted by yellow dots. Numbers denote specific cave identities related to numbers assigned by Gavish-Regev et al. [19].

In each cave ecological zone, as well as outside the cave, temperature and illumination were measured. The temperature was measured using a PicoLite 16-K and a single-trip USB temperature logger (FOURTEC, Rosh Ha'Ayin, Israel), with measurements taken once every hour for 74–77 days. The illumination was recorded at the time of each survey using an Extech 401025 Lux Light Meter (Extech, Nashua, NH, USA). The light meter was positioned on the ground until the reading stabilised for a minimum of 1 min. Illumination measurements inside caves ranged between 0 and 420 lux, while measurements outside caves ranged between 60 and 70,000 lux. Temperature and illuminance were also measured outside the caves. Cave length was estimated from cave maps when available (via the Israel Cave Research Centre), or in the field by measuring the distance from the cave opening to the darkest region of the cave accessible during the surveys. Elevation and geological data were provided by the GIS (geographic information system) Centre at the Hebrew University of Jerusalem. Each of the 35 caves was sampled twice according to a specific protocol during 2014 (6 March–6 April 2014 and 22 May–22 June 2014). Due to differences in cave morphology (including microhabitat, fractal shape of the substrates, size, and volume), we standardised our sampling effort by time. Our protocol included a 20 min thorough visual search by one of three experienced arachnologists in 3–10 m long zones using headlamps and UV lights in each of the ecological zones of each cave. In the first visit to each cave, most of the arachnids observed were collected by hand for further

identification in the lab. In further visits to each cave, species considered sensitive or common were identified to the species level in situ, without the need to collect specimens (see Gavish-Regev et al. (2021) for additional information and species lists [19]). Voucher specimens were deposited in the National Arachnid Collection in the National Natural History Collections of The Hebrew University of Jerusalem (NNHC, HUJI).

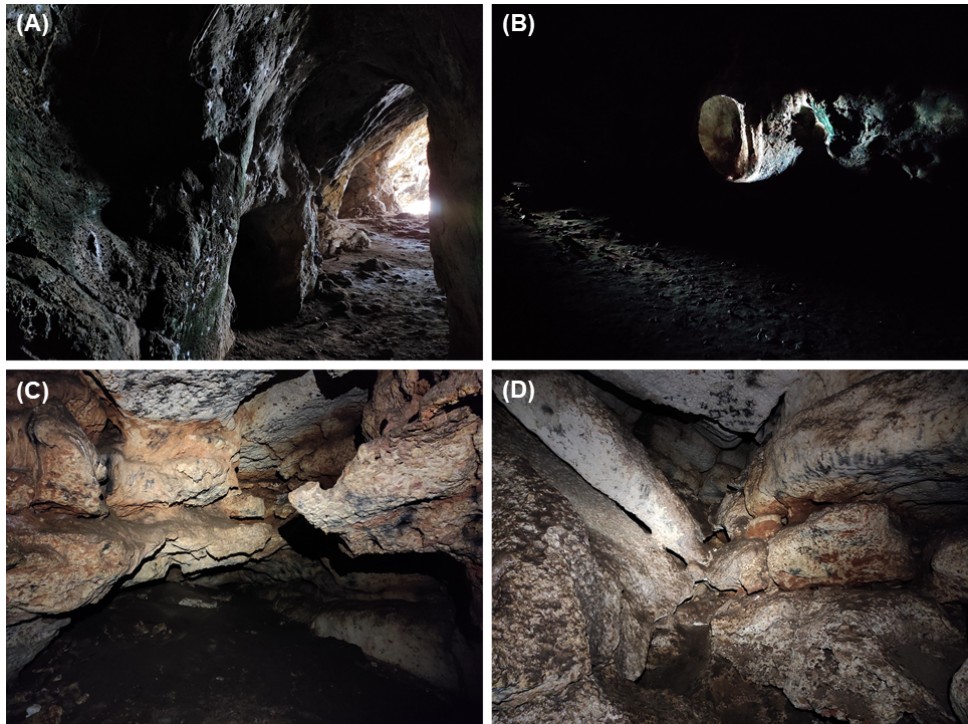

**Figure 2.** Examples of the distinct ecological zones of a cave (Ornit cave in the Karmel mountain ridge): (**A**) The cave entrance visible from the twilight zone chamber. (**B**) The twilight zone visible from the back of the twilight zone chamber. (**C**,**D**) Example photographs of the dark zone taken with flash. Photos taken by Shlomi Aharon.

## 2.2. Statistical Analysis

All analyses were conducted in R v4.0.3 [21]. To ascertain the diversity of the ecological zones of the caves surveyed, and the completeness of those surveys, coverage-based rarefaction and extrapolation were carried out, and Hill diversity was calculated as a robust estimate of species diversity [22,23]. This was performed using the "iNEXT" package, with species represented by their frequency of occurrence across samples [22,24]. These Hill numbers include the three most widely-used diversity measures—species richness, Shannon diversity, and Simpson diversity—each varying in their diversity order, $q$ ($q = 0$, 1, and 2, respectively), which indicates the sensitivity of the measure to relative abundance [24]. Species richness counts species equally, irrespective of their relative abundance; Shannon diversity counts individuals equally, thus representing species proportional to their abundance; and Simpson diversity gives greater weight to the dominant species in the assemblage. These were visually represented by plotting the cumulative diversity (according to all three indices) against the number of detections (occurrences), the cumulative sample coverage (i.e., completeness of sampling) against the number of detections, and the cumulative diversity against the sample coverage. Visual inspection of these figures effectively facilitates the drawing of conclusions regarding the suitability of sampling efforts, and how diversity is structured in cave zones.

Spider assemblages were compared between cave zones using multivariate generalised linear models (MGLMs) via "manyglm" in the "mvabund" package [20], with a Poisson error family and Monte Carlo resampling. To account for the presence of multiple

ecological zones in the same cave, any entrance zones in caves containing twilight zones, and any twilight zones in caves containing dark zones, were removed for this analysis; this left 10 entrance zones, 13 twilight zones, and 11 dark zones, all from different caves. Model independent variables included cave zone (entrance, twilight, or dark), zone minimum temperature, cave elevation, geographical region, and all two-way interactions between cave zone and the remaining variables. Geographical region was included, since it was found to be an important factor in spider assemblage composition in a previous study, and may disentangle any regional effects of zone assemblage differences [19]. An alternative analysis, presented in Appendix A, was carried out without excluding zones from the same cave. This enables the reader to inspect results from the complete dataset. This analysis is less conservative and should be viewed with caution; since we were unable to include cave identity as a random effect in these models, we could not discount autocorrelation in the data. The analysis, however, is supportive of the findings of the more conservative analysis presented in the main text.

Coarse differences between assemblages were visualised by non-metric multidimensional scaling (NMDS) via metaMDS in the "vegan" package [25], with Bray–Curtis distance in 2 dimensions and 999 tries. For visualisation of the effect of categorical variables against the NMDS, spider plots were created using "ordispider" with "ggplot2" [26]. For visualisation of the effect of continuous variables against the NMDS, surf plots were created with scaled coloured contours using "ordisurf" with "ggplot2".

Pairwise co-occurrence analysis was carried out to identify spider species that occurred together more, or less, than expected by chance. The "cooccur" function in the "cooccur" package [27] was used to calculate the expected frequencies of co-occurrence between each pair of taxa via null models, which were then compared against the observed patterns of co-occurrence to identify deviations from random. Nestedness—the ordered loss of species—was assessed across cave zone spider assemblages using the binary-matrix nestedness temperature calculator (BINMATNEST; [28,29]). A binary presence–absence matrix was used to calculate "temperature" (0–100 °C; the deviation from perfect nestedness, represented by 100 °C). The nestedness of the binary matrix was compared against 100 matrices generated randomly using null models [30].

## 3. Results

### 3.1. Spider Assemblage Diversity in Cave Zones in the Southern Levant

Across the surveys, a total of 1054 spiders were collected or identified in the field, belonging to 62 species and morphospecies across 38 genera and 22 families. Many species were found across all three cave zones, but some in only one or two (Figure 3). The cave assemblages were highly diverse (Hill richness = $63.00 \pm 7.79$; Hill–Shannon diversity = $32.21 \pm 4.92$; Hill–Simpson diversity = $19.53 \pm 4.38$; Figure 4) and largely composed of rare species, with many species found only in a single cave. The surveys identified an estimated 89.57% ($\pm$ 2.8%) of the total spider diversity in caves (Figure 4). Limited nestedness was observed across cave zones (3.98 °C, vs. $16.79 \pm 2.79$ °C in the null models; $p < 0.05$).

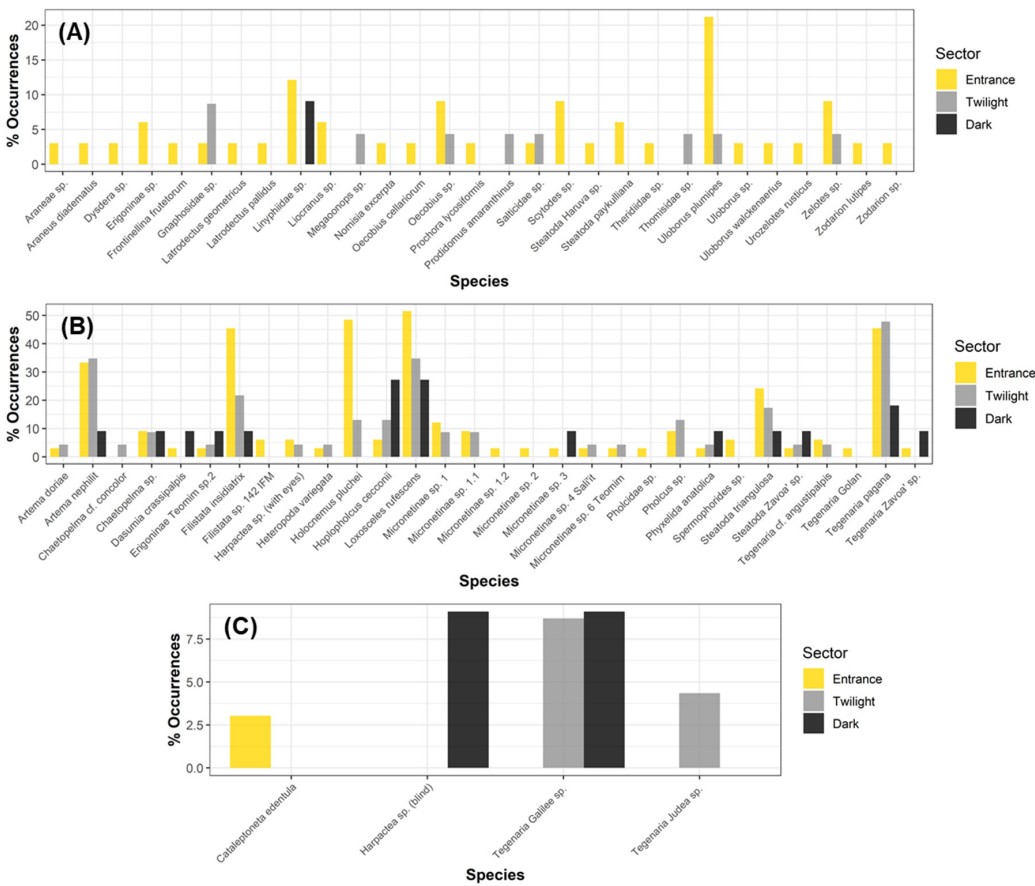

**Figure 3.** Percentage of surveys in which (**A**) accidental, (**B**) troglophile, and (**C**) troglobite species were identified across cave zones. Dark, twilight, and entrance abundances are denoted by black, grey and yellow, respectively. Occurrence is calculated as the percentage of surveys of a zone type that found at least one individual of each species.

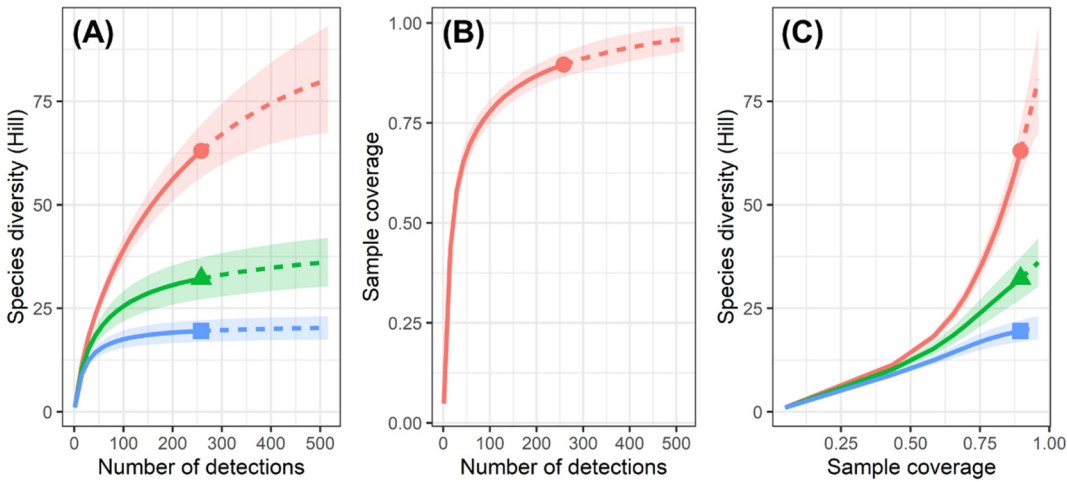

**Figure 4.** Diversity calculated from cave surveys and the associated sample coverage: (**A**) Species diversity per number of spider detections (occurrences). (**B**) Sample coverage per number of spider detections (occurrences). (**C**) Species diversity by sample coverage. Red lines with terminal circles, green lines with terminal triangles, and blue lines with terminal squares denote Hill-richness, Hill–Shannon diversity, and Hill–Simpson diversity, respectively. Solid lines represent observed diversity, and dashed lines represent extrapolated diversity. Light zones surrounding lines denote 95 % confidence intervals.

### 3.2. Zone Assemblage Comparison

Specific spider assemblages were significantly related to cave zones (MGLM: Dev = 426.0, *d.f.* = 31, *p* = 0.001; Figure 5), but also to the interaction between zone and minimum temperature (MGLM: Dev = 124.8, *d.f.* = 21, *p* = 0.001), elevation (MGLM: Dev = 68.3, *d.f.* = 19, *p* = 0.001), and geographical region (MGLM: Dev = 109.5, *d.f.* = 23, *p* = 0.001). Specific cave zone assemblages were also significantly associated with differences in minimum temperature (MGLM: Dev = 254.5, *d.f.* = 30, *p* = 0.001; Figure 6), elevation (MGLM: Dev = 229.5, *d.f.* = 29, *p* = 0.001; Figure 6), and geographical region (MGLM: Dev = 397.2, *d.f.* = 27, *p* = 0.001). Fourteen species were significantly associated with cave zones or interactions between cave zone and other variables; of these, six spider species were significantly associated with interactions between cave zone and temperature, cave length, or geographical region (Table 1, Figure 7). A further five spider species were significantly associated with minimum temperature, elevation, or geographical region (Table S1, in supplementary material). Specifically, *Artema nephilit* (Aharon, Huber, and Gavish-Regev, 2017), *Filistata insidiatrix* (Forsskål, 1775) (interacting with minimum temperature), and *Tegenaria pagana* (C.L. Koch, 1840) (interacting with geographical region, minimum temperature, and elevation) were almost equally common in both twilight and entrance zones; *Filistata* sp., *Holocnemus pluchei* (Scopoli, 1763), *Loxosceles rufescens* (Dufour, 1820) (interacting with geographical region and minimum temperature), Micronetinae sp. (interacting with minimum temperature), *Oecobius* sp., *Steatoda triangulosa* (Walckenaer, 1802) (interacting with geographical region and minimum temperature), and *Tegenaria angustipalpis* (Levy, 1996) in entrance zones; *Pholcus* sp. in twilight zones; *Tegenaria* sp. from the Galilee (interacting with minimum temperature) almost equally in both twilight and dark zones; and *Hoplopholcus cecconii* (Kulczynski, 1908) and *Tegenaria* sp. from Zavoa cave in dark zones (Figure 7).

**Table 1.** Significant univariate MGLM results for the 14 species with significant associations with cave zones, or with interactions between cave zones and other variables; deviance and probability are given for each. Other significant variables are listed, as well as interactions between zone and those variables. For the other associations, the nature of the association is given as + or − for positive or negative associations, respectively, and "N", "C", and "S" are given for prevalence in north, central, or southern geographical regions, respectively. The "category" column denotes whether that species is troglobite, troglophile, or accidental (see Gavish-Regev et al., 2021 [19] for the assignment of species to categories). The "dominant zone" column denotes the ecological zones(s) in which the species was mostly commonly found. The abundance of these spiders across different cave zones is visualised in Figure 6. For spiders with significant associations with interactions between cave zone and other variables, but not with the cave zones themselves (*n* = 3), the deviance and probability are given in italics.

| Species | Category | Dominant Zone | Dev | *p* | Other Associations | Interactions |
|---|---|---|---|---|---|---|
| *Artema nephilit* | Troglophile | Twilight/Entrance | 34.520 | 0.001 | −Elevation (Dev = 40.978, *p* = 0.001) | - |
| *Filistata insidiatrix* | Troglophile | Entrance/Twilight | 15.973 | 0.005 | −Temperature (Dev = 33.296, *p* = 0.001) −Elevation (Dev = 10.758, *p* = 0.027) N Region (Dev = 29.727, *p* = 0.001) | Zone/Temperature (Dev = 19.662, *p* = 0.002) |
| *Filistata* sp. | Troglophile | Entrance | 53.846 | 0.001 | C Region (Dev = 69.669, *p* = 0.001) | - |

**Table 1.** *Cont.*

| Species | Category | Dominant Zone | Dev | p | Other Associations | Interactions |
|---|---|---|---|---|---|---|
| *Holocnemus pluchei* | Troglophile | Entrance | 36.565 | 0.001 | −Temperature (Dev = 35.308, *p* = 0.001) −Elevation (Dev = 30.028, *p* = 0.001) | - |
| *Hoplopholcus cecconii* | Troglophile | Dark | 24.826 | 0.001 | + Elevation (Dev = 17.750, *p* = 0.004) N Region (Dev = 12.346, *p* = 0.042) | - |
| *Loxosceles rufescens* | Troglophile | Entrance | *8.525* | *0.129* | N Region (Dev = 52.706, *p* = 0.001) | Zone/Region (Dev = 25.701, *p* = 0.001) Zone/Temperature (Dev = 19.662, *p* = 0.002) |
| *Micronetinae* sp. (1) | Troglophile | Entrance | *8.585* | *0.129* | - | Zone/Temperature (Dev = 12.237, *p* = 0.008) |
| *Oecobius* sp. | Accidental | Entrance | 15.224 | 0.008 | + Elevation (Dev = 21.825, *p* = 0.001) | - |
| *Pholcus* sp. | Troglophile | Twilight | 11.537 | 0.032 | C Region (Dev = 31.480, *p* = 0.001) | - |
| *Steatoda triangulosa* | Troglophile | Entrance | 21.631 | 0.001 | - | Zone/Region (Dev = 17.309, *p* = 0.002) Zone/Temperature (Dev = 13.875, *p* = 0.006) |
| *Tegenaria angustipalpis* | Troglophile | Entrance | 17.133 | 0.003 | + Elevation (Dev = 11.189, *p* = 0.027) N Region (Dev = 18.917, *p* = 0.001) | - |
| *Tegenaria pagana* | Troglophile | Entrance/Twilight | *3.359* | *0.858* | − Elevation (Dev = 19.611, *p* = 0.003) N/C Region (Dev = 53.838, *p* = 0.001) | Zone/Region (Dev = 19.935, *p* = 0.001) Zone/Temperature (Dev = 27.866, *p* = 0.001) Zone/Elevation (Dev = 37.165, *p* = 0.001) |
| *Tegenaria* sp. from Galilee caves | Troglobite | Twilight/Dark | 19.865 | 0.001 | + Elevation (Dev = 16.046, *p* = 0.006) N Region (Dev = 27.424, *p* = 0.001) | Zone/Temperature (Dev = 13.034, *p* = 0.006) |
| *Tegenaria* sp. from Zavoa cave | Troglophile | Dark | 38.368 | 0.001 | + Temperature (Dev = 81.525, *p* = 0.001) | - |

### 3.3. Co-Occurrence of Spiders in Cave Zones

Of the 1953 possible pairwise species co-occurrence combinations, 1872 (95.85%) were not carried forward for co-occurrence analysis, since they were expected to co-occur less than once. Of the 81 pairs analysed, 68 co-occurred randomly, and 13 (16%) non-randomly (Figure 8, Figure S2, Table 2). Of these non-randomly co-occurring pairs, 11 co-occurred more than expected by chance—*Chaetopelma* sp. with *Loxosceles rufescens*, *F. insidiatrix* with *H. pluchei*, *F. insidiatrix* with *L. rufescens*, *F. insidiatrix* with *Tegenaria pagana*, *L. rufescens* with *H. pluchei*, *H. pluchei* with Micronetinae sp. 1, *H. pluchei* with *Steatoda triangulosa*, *H. pluchei* with *U. plumipes*, *L. rufescens* with Micronetinae sp. 1, *L. rufescens* with *T. pagana*, and *Pholcus* sp. with *T. pagana*—and 2 less than expected by chance: *A. nephilit* with *Hoplopholcus cecconii*, and *Artema nephilit* with *S. triangulosa* (Table 2, Figure 8).

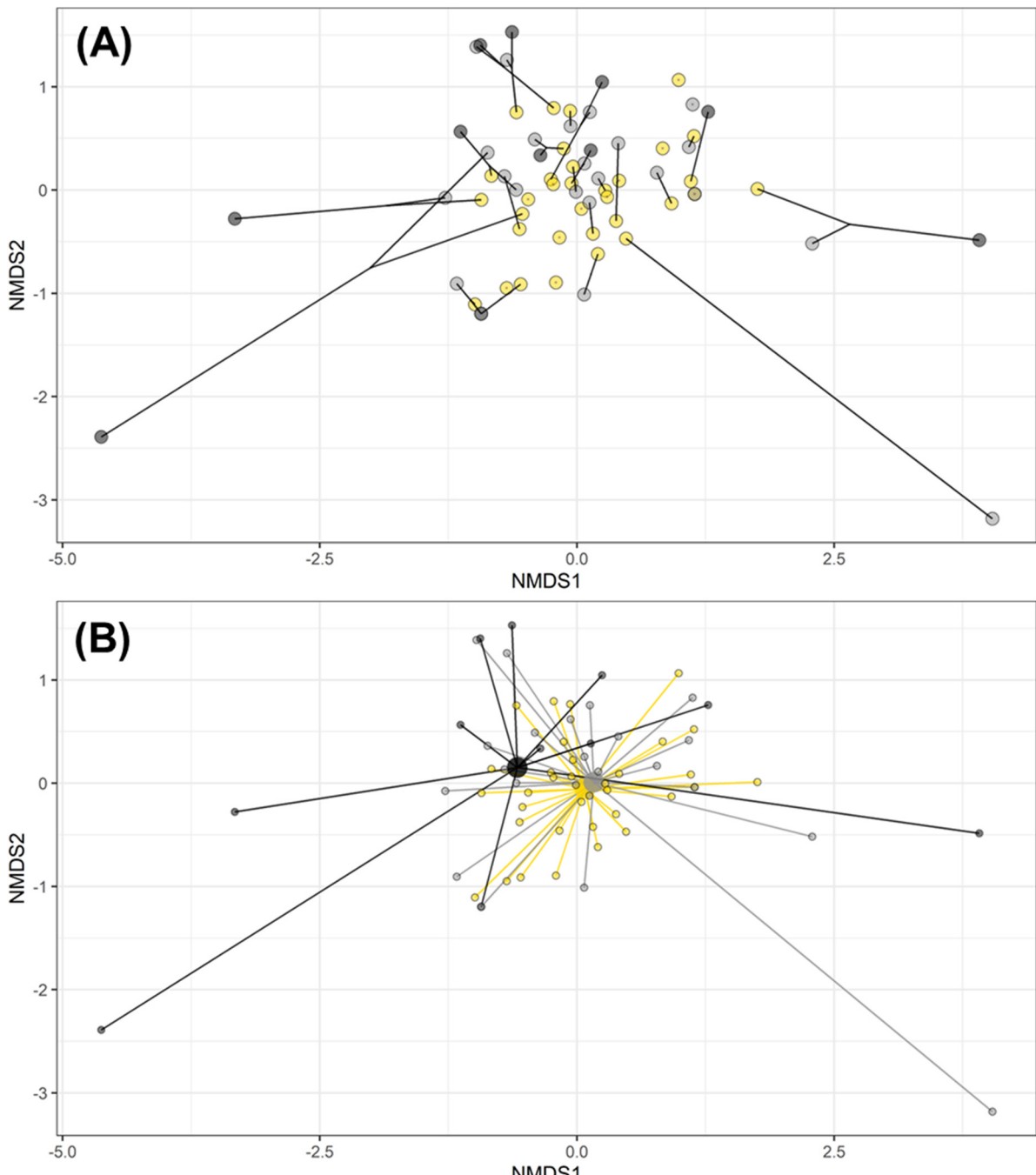

**Figure 5.** Spider plots of assemblages by cave zone derived from non-metric multidimensional scaling. Colours denote cave zones (black, grey, and yellow denoting dark, twilight, and entrance zones, respectively). Axes represent a two-dimensional variation in spider assemblages. Each smaller point in both images represents a single assemblage present in a specific cave and a specific zone, with the distance between them indicating their dissimilarity (i.e., proximate points are similar, distant points are dissimilar). Average species coordinates are overlaid in Figure S1. Stress = 0.1002019. (**A**) Joined points belong to the same cave; these are typically more similar (i.e., points are closer) than those of other caves, with the exception of many of the dark zone assemblages (generally around the outside of the central cluster of points), which show substantial variation. (**B**) Points are joined by the centroids of assemblages in each group (larger points = mean coordinates in that group); a high degree of overlap is observable, particularly between twilight and entrance assemblages, but dark zone assemblages are more distinct and highly variable.

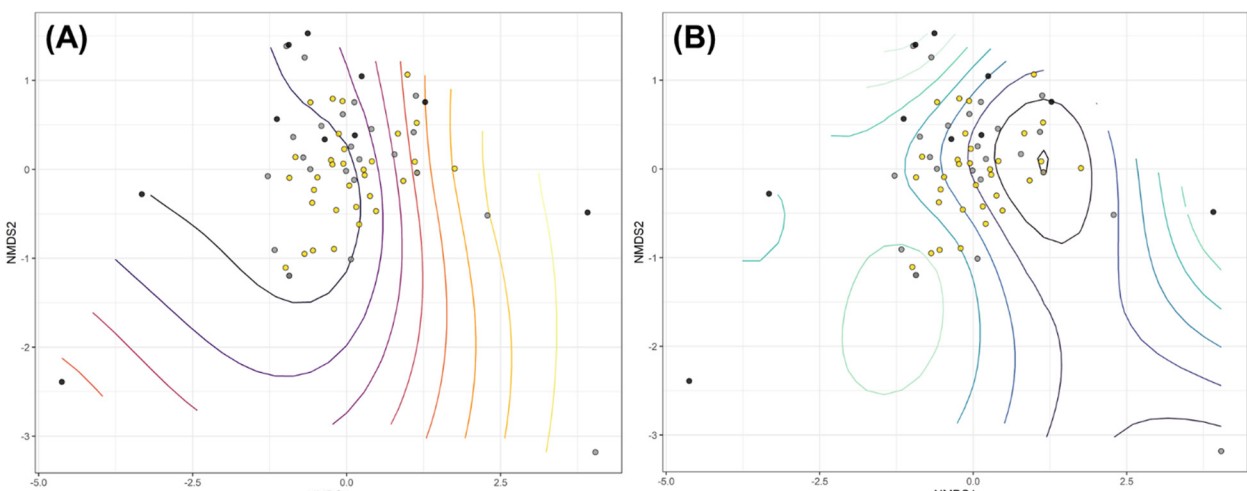

**Figure 6.** Surf plots of assemblages by (**A**) minimum cave temperature and (**B**) cave elevation, both derived from non-metric multidimensional scaling. Each point represents a single assemblage present in a specific zone of a specific cave, with the distance between them indicating their dissimilarity (i.e., proximate points are similar, distant points are dissimilar). Point colours denote cave zones (black, grey, and yellow denoting dark, twilight, and entrance zones, respectively). Contour colours denote (**A**) temperature (purple–yellow denoting low–high temperatures) and (**B**) elevation (dark–light blue denoting low–high elevation). Axes represent a two-dimensional variation in spider assemblages. Most of the assemblages that existed in cooler caves appear to be more similar than those in warmer caves. A complex nonlinear trend is observable between assemblage composition and elevation, with assemblages in intermediate-elevation caves appearing more similar. Average species coordinates are overlaid in the spider plot constructed with the same sample coordinates in Figure S1. Stress = 0.1002019.

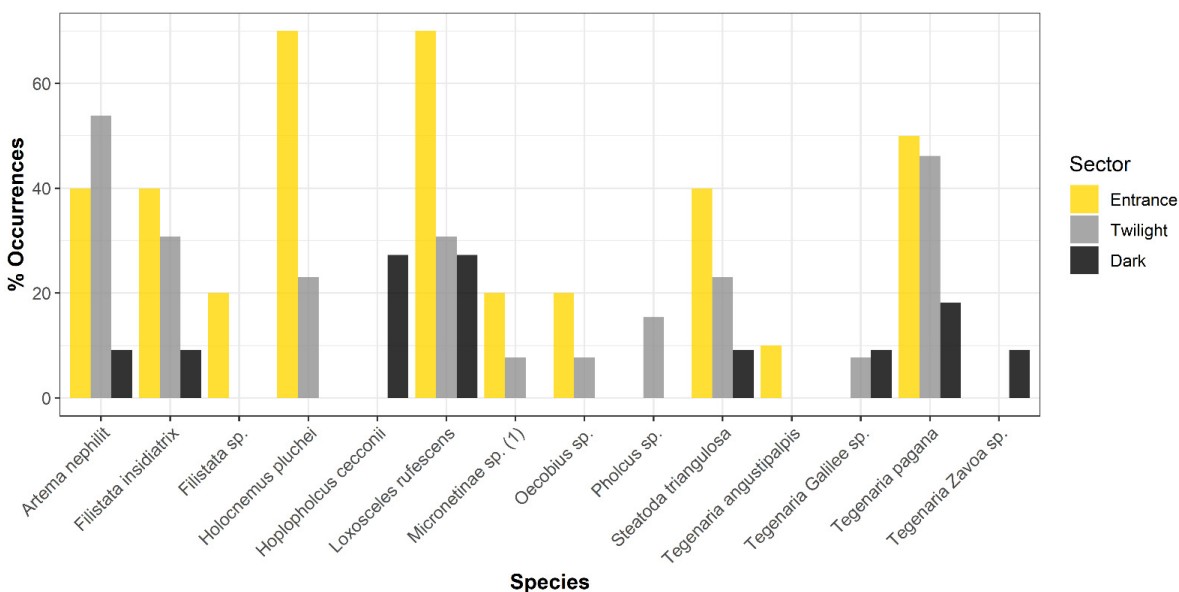

**Figure 7.** Percentage of surveys in which the 14 species with significant associations with cave zones were identified across cave zones. Only those cave zones included in the MGLM are represented here (i.e., some species may have been found in other zones (i.e., *Hoplopholcus cecconii* (entrance, twilight), *Pholcus* sp. (entrance), *Tegenaria angustipalpis* (twilight), and *T.* sp. from Zavoa cave (entrance)), but due to subsampling of the data and the use of only one zone per cave for the MGLM, these data are absent from this graph (see Figure 3 and Appendix A for the complete dataset, and Materials and Methods for more details regarding the MGLM). Dark, twilight, and entrance zone abundances are denoted by black, grey, and yellow, respectively. Occurrence is calculated as the percentage of surveys of a zone type in which at least one individual of each species was found.

**Table 2.** Species co-occurrence across cave zones. The probability corresponds to the probability that the respective species co-occur more, or less, than expected (listed as "relationship", with "+" and "−" denoting positive and negative co-occurrences, respectively). Negatively co-occurring species are in bold.

| Species 1 | Species 2 | Expected Co-Occurrences | Observed Co-Occurrences | Co-Occurrence Relationship | *p*-Value |
|---|---|---|---|---|---|
| *Artema nephilit* | *Hoplopholcus cecconii* | 2.4 | 0 | − | 0.048 |
| *Artema nephilit* | *Steatoda triangulosa* | 3.9 | 1 | − | 0.047 |
| *Chaetopelma* sp. | *Loxosceles rufescens* | 2.5 | 6 | + | 0.004 |
| *Filistata insidiatrix* | *Holocnemus pluchei* | 6 | 11 | + | 0.005 |
| *Filistata insidiatrix* | *Loxosceles rufescens* | 8.8 | 15 | + | 0.001 |
| *Filistata insidiatrix* | *Tegenaria pagana* | 8.8 | 14 | + | 0.006 |
| *Holocnemus pluchei* | *Loxosceles rufescens* | 7.9 | 12 | + | 0.026 |
| *Holocnemus pluchei* | Micronetinae sp. 1 | 1.7 | 4 | + | 0.05 |
| *Holocnemus pluchei* | *Steatoda triangulosa* | 3.7 | 7 | + | 0.03 |
| *Holocnemus pluchei* | *Uloborus plumipes* | 2.3 | 5 | + | 0.036 |
| *Loxosceles rufescens* | Micronetinae sp. 1 | 2.5 | 5 | + | 0.042 |
| *Loxosceles rufescens* | *Tegenaria pagana* | 11.7 | 18 | + | 0.002 |
| *Pholcus* sp. | *Tegenaria pagana* | 2.5 | 5 | + | 0.042 |

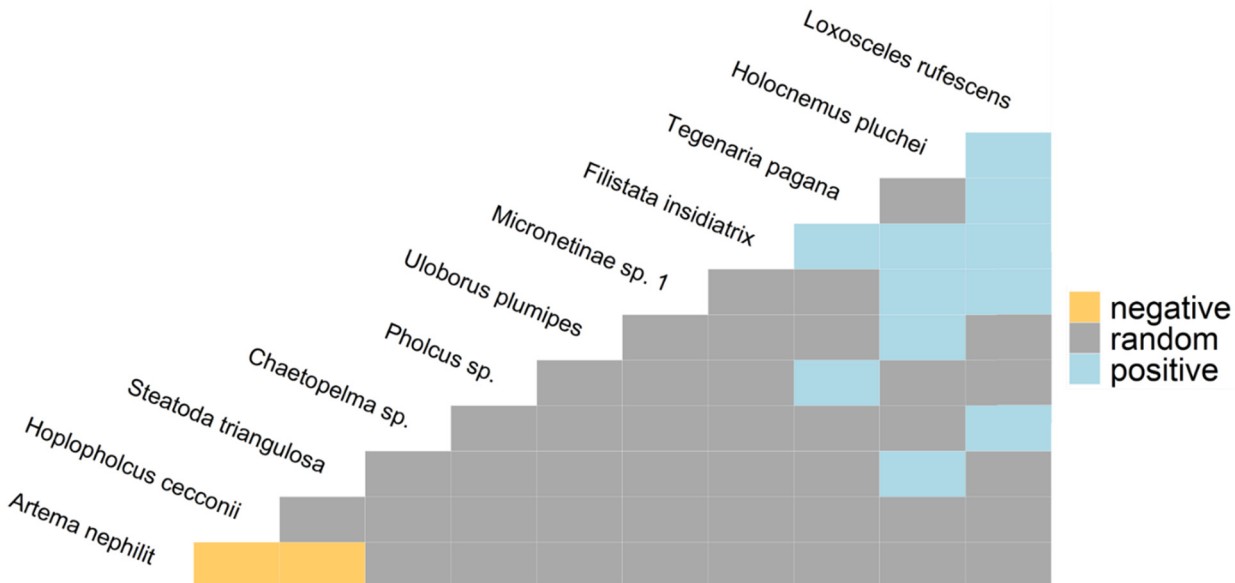

**Figure 8.** Species co-occurrence matrix for those species identified from cave spider assemblages from distinct cave zones. Yellow, grey, and blue points denote significantly negative, random, and significantly positive co-occurrences, respectively.

## 4. Discussion

We have shown that the different ecological zones of caves host unique assemblages of spiders, including many rare species. The diverse cave spider assemblages within these environments are structured by a high degree of endemicity, with many species occurring only in a single cave or cave zone. The fact that this study identified an estimated 89.57% of the total assemblages in these caves, based on rarefied sample coverage, was considered promising and unsurprising given the endemicity of many of the extant fauna; each additional cave sampled is likely to host a unique assemblage, possibly including its own endemic species—especially where those caves contain multiple distinct ecological zones. Indeed, such caves have recently yielded taxa new to Israel and new to science [31–34].

Cave assemblages show a graded pattern of dissimilarity, with dark and entrance assemblages appearing the most dissimilar, and twilight assemblages existing as an intermediate. The relative similarity of entrance assemblages between different caves, compared with the dissimilarity of different dark assemblages (as exemplified in Figure 5), suggests that dark zone assemblage composition is not directly derived from the species pool at the cave entrance, but may have diverged from ancestral entrance assemblages over time. This is supported by a low nestedness of the assemblages within the caves, contrary to our original hypothesis. These patterns of dissimilarity and low nestedness may be due to the existence of relict species in some of the caves sampled, or to speciation within caves as a consequence of low gene flow [1]. Weak gene flow could result in a transition from entrance assemblage species to distinct dark zone troglobite species through adaptive radiation [1,35]. Following the adaptive shift hypothesis (ASH; [36]), troglobitic dark zone spider assemblages could have developed from entrance and twilight assemblages, but differentiated from them over time through these speciation events. These assemblages may have included species with pre-adaptations to caves, such as those adapted to shallow depressions or hollows [37], thus facilitating further adaptation to dark zones. Following the climatic relict hypothesis (CRH), dark zone assemblages may include ancestral or relict species that did not possess pre-adaptations, but survived in caves after changes in climatic conditions rendered the epigean habitat unsuitable for them [38]. The two hypotheses are not mutually exclusive, however, and our study—an ecological snapshot in time—was not designed to test these hypotheses. Further study is required in order to fully elucidate these evolutionary processes.

Our models explaining the assemblage structures in different caves revealed interactions between ecological zone and other variables such as minimum temperature, elevation, and region, which may reflect the particular conditions found in caves hosting multiple zones. The differences in assemblage structure between zones were mostly affected by those taxa individually associated with different zones (highlighted in Table 1). These species belong to just a few families (Agelenidae, Filistatidae, Linyphiidae, Pholcidae, Sicariidae, and Theridiidae). The two most common families among these—Pholcidae and Agelenidae—had consistent and generally opposite cave zone occupancies. Pholcidae (troglophile species) were most often prevalent in the entrance, with a consistent presence in the twilight zone. One species (*Hoplopholcus cecconii*), however, occurred often in the dark zone, as seen in the data used in the MGLM (Table 1, but for complete data, see Figure 3 and Appendix A). When considering these additional observations, *H. cecconii* was most abundantly found in the twilight and entrance zones (Appendix A), indicating that this species was in fact found in all three zones, but of the same caves, and was exclusively found in caves large enough to contain all three ecological zones.

Agelenidae appeared across all three zones, with four species that differed in abundance between zones (*T. angustipalpis, T. pagana*, and two undescribed species found in the Galilee caves and in the Zavoa cave) and occurred in different permutations of cave zones, indicating a high degree of habitat generalism. Most of the species that differed significantly in their zone occupancy were troglophiles. In the MGLM, only three of these troglophiles were not found in the cave entrance: *Pholcus* sp., *Hoplopholcus cecconii*, and *Tegenaria* sp. Zavoa. All of these species, however, were found in entrance zones, which were excluded from the MGLM because they were from the same cave as a zone already represented in the model (as explained in the Materials and Methods section and Appendix A). The only troglobite species associated with a specific cave zone was *Tegenaria* sp. Galilee, found exclusively in the dark and twilight zones. Congeneric species such as *Tegenaria pagana* occupied both the cave entrance and twilight zones, indicating that zone occupancy is not taxon-specific, with species within a genus differing in their adaptation to cave systems. The only accidental species with significant association with a specific cave zone (*Oecobius* sp.) also occurred almost exclusively in the entrance, suggesting that this accidental species may lack the pre-adaptations necessary to thrive in twilight and dark zones.

The 13 co-occurrence relationships identified as deviations from random highlight some consistencies in the caves' regional spider assemblage structuring. The positive co-occurrence relationships prominently include the most common species—namely, *Holocnemus pluchei* (5/13 significant co-occurrences, found in 28% of zone surveys) and *Filistata insidiatrix* (3/13 significant co-occurrences, found in 31% of zone surveys). These co-occurrences may have been more frequent than expected simply because these species are widespread and common representatives of the regional cave assemblages in all but the dark zones.

Only two co-occurrences were identified as being significantly less common than expected: *Artema nephilit* with *Hoplopholcus cecconii*, and *A. nephilit* with *Steatoda triangulosa*. *Artema nephilit*, which was encountered in 30% of zone surveys, was not present in any positive co-occurrence relationships, despite being in both negative co-occurrence relationships. *Artema nephilit* was found mainly in warmer caves (Appendix A), and was often among a very small number of species found in hot and dry caves (Gavish-Regev, personal observations). The specific negative co-occurrence of *A. nephilit* and *Hoplopholcus cecconii* is probably a result of different habitat requirements or, as highlighted by the MGLM results (Table 1), differences in elevation. Additionally, *A. nephilit* was found mainly in the Rift Valley to the east, whereas *H. cecconii* occurred mainly toward the Mediterranean region in the west; this would not be identified by the MGLMs, despite their inclusion of region, since the analysis was focused on a north/central/south division. Where the distributions of *A. nephilit* and *H. cecconii* do overlap, in the Karmel, *A. nephilit* was found exclusively in the warmer, south-facing cave, while *H. cecconii* was found only in the cooler, north-facing cave, just a few meters away.

The distribution of spiders in caves can be affected by niche dynamics and competition for resources, with dominant species forcing species with pre-adaptations away from entrance zones and into the relatively prey-poor twilight and dark zones [7]. This "step back effect" may be a significant driver of diversity in the innermost regions of caves [4]. Such effects could theoretically be responsible for the negative co-occurrence relationships between species such as *A. nephilit* and *Steatoda triangulosa*; although in this case these two species were not present together in the same caves, they occupy a similar spatial—and likely trophic—niche in the caves, and their negative association could result from competitive exclusion of one another. A similar process may have occurred in Pholcidae, since most caves contain only a single species of Pholcidae [39]. Broader analyses of cave spider prey communities, interaction networks, and trophic ecology may further disentangle additional spatial patterns in, and drivers of, spider cave occupancy.

## 5. Conclusions

Cave ecological zones host divergent yet diverse assemblages of spiders. This diversity largely reflects an adaptive process of speciation, from relatively unspecialised and widely distributed entrance zone assemblages, to highly specialised dark zone assemblages comprising many endemic species. Our results indicate that differences in assemblages between cave zones are further affected by temperature, elevation, and geographical region, but the patterns of co-occurrence in these assemblages also suggest that competition is a driver of cave spider spatial dynamics. The high degree of endemicity in these systems lends credence to their importance as hotspots of biodiversity, but their relative climatic stability does not render them impervious to the effects of climate change and other drivers of species loss [40]. A greater understanding of the drivers of diversity in cave spider populations is necessary in order to ensure sufficient knowledge to protect these important and unique contributors to the global arachnofauna.

**Supplementary Materials:** The following are available online at https://www.mdpi.com/article/10.3390/d13110576/s1, Figure S1: Spider plot of communities by cave zone, derived from non-metric multidimensional scaling. Colours denote cave zones (black, grey, and yellow denoting dark, twilight, and entrance zones, respectively). Axes represent a two-dimensional variation in spider communities. Each smaller point represents a single community present in a distinct cave

zone, joined by the centroids of communities in each group (larger nodes = mean coordinates in that group), with the distance between them indicating their dissimilarity (i.e., proximate points are similar, distant points are dissimilar). A high degree of overlap is observable, particularly between twilight and entrance communities, but dark communities are more distinct and highly variable. Stress = 0.1002019; Figure S2: Spider species co-occurrence across cave zones. Yellow, grey, and blue points denote significantly negative, random, and significantly positive co-occurrences, respectively; Table S1: Significant univariate MGLM results for species with significant associations with model independent variables not including cave zone or interactions between cave zone and other variables; deviance and probability are given for each. The nature of the association with temperature or elevation is given as "+" or "−" for positive or negative associations, respectively, and "N", "C", and "S" are given for prevalence in north, central, or southern geographical regions, respectively. The "category" column denotes whether a species is troglobite, troglophile, or accidental (see Gavish-Regev et al., 2021 for the assignment of species to categories [19]).

**Author Contributions:** Conceptualisation, J.P.C., E.G.-R., S.A., and Y.L.; methodology, E.G.-R., S.A., M.S., and Y.L.; analysis, J.P.C., and E.G.-R.; investigation, S.A., I.A.S., and E.G.-R.; resources, E.G.-R., Y.L., and M.S.; data curation, E.G.-R., S.A., and J.P.C.; writing—original draft preparation, J.P.C., and E.G.-R.; writing—review and editing, J.P.C., E.G.-R., S.A., I.A.S., M.S., and Y.L.; project administration, E.G.-R.; visualisation, J.P.C., S.A., and E.G.-R.; supervision, E.G.-R., M.S., and Y.L.; funding acquisition, E.G.-R., and Y.L. All authors have read and agreed to the published version of the manuscript.

**Funding:** This research was funded by the Israel Taxonomy Initiative (ITI) biodiversity survey grant to E.G.-R. and Y.L., ISF grant (2656/20) to E.G.-R., and ITI and Ben-Gurion University fellowships to S.A.

**Institutional Review Board Statement:** Permit granted by the Israel Nature and Park Authority. No further need for ethical approval was required for invertebrate research.

**Informed Consent Statement:** Not applicable.

**Data Availability Statement:** The data presented in this study are publicly available via Zenodo: https://doi.org/10.5281/zenodo.5590782.

**Acknowledgments:** This research was conducted with collection permits 2013/40027, 2013/40085, and 2014/40313 from the Israel National Parks Authority. We thank M. Isaia and S. Mammola for discussions on the arachnid cave survey; A. Frumkin, B. Langford, and E. Cohen (Israel Cave Research Centre) for assistance with cave data and in the field; A. Ben-Nun for assistance with GIS and maps (GIS Centre at the Hebrew University of Jerusalem); and L.E. Drake for advice on the multivariate modelling.

**Conflicts of Interest:** The authors declare no conflict of interest.

## Appendix A

In the multivariate generalized linear model (MGLM) analyses presented in the main text, any entrance zones in caves containing twilight zones, and any twilight zones in caves containing dark zones, were removed to account for the presence of multiple ecological zones in the same cave. This left 10 entrance zones, 13 twilight zones, and 11 dark zones, all from different caves (total = 34). This is approximately half of the total number of cave zones surveyed and, thus, represents a reduction in statistical power, as well as the loss of many observations. An alternative analysis was thus carried out without excluding zones from the same cave, and is presented herein, but should be viewed with caution and scepticism. Since MGLMs do not allow the inclusion of random factors—in this case, cave identity—this analysis will be affected by the autocorrelation of the data collected from the same caves. We have decided to present it nevertheless, in order to highlight the overall consistency of the analyses, but also the effects of some of the additional observations (e.g., the large number of observations of *Pholcus* sp. in entrance zones).

### Appendix A.1. Materials and Methods

Spider assemblages were compared between cave zones using multivariate generalized linear models (MGLMs) via "manyglm" in the "mvabund" package [20], with a

Poisson error family and Monte Carlo resampling. Model independent variables included cave zone (entrance, twilight, or dark), zone minimum temperature, cave elevation, geographical region, and all two-way interactions between cave zone and the other variables.

*Appendix A.2. Results*

Specific spider assemblages were significantly related to cave zones (MGLM: Dev = 476.4, *d.f.* = 64, *p* = 0.001), and interactions between zone and temperature (MGLM: Dev = 113.4, *d.f.* = 51, *p* = 0.001), elevation (MGLM: Dev = 53.0, *d.f.* = 49, *p* = 0.007), and geographical region (MGLM: Dev = 818.0, *d.f.* = 59, *p* = 0.001). Ten species were found to have significant associations with cave zones; three of these species were also significantly associated with temperature and geographical region (Table A1, Figure A1). Specifically, *Artema nephilit* (Aharon, Huber, and Gavish-Regev, 2017) and *Pholcus* sp. were most common almost equally in both twilight and entrance zones; *Filistata insidiatrix* (Forsskål, 1775) (depending on an interaction between zone and geographical region), *Filistata* sp., *Holocnemus pluchei* (Scopoli, 1763) (depending on an interaction between zone and geographical region), and *Uloborus plumipes* (Lucas, 1846) in entrance zones; *Tegenaria* sp. from cave in Judea in twilight zones; *Tegenaria* sp. from caves in the Galilee (depending on an interaction between zone and temperature) almost equally in both twilight and dark zones; and *Steatoda* sp. from Zavoa cave and *Tegenaria* sp. from Zavoa cave in dark zones (Figure A1).

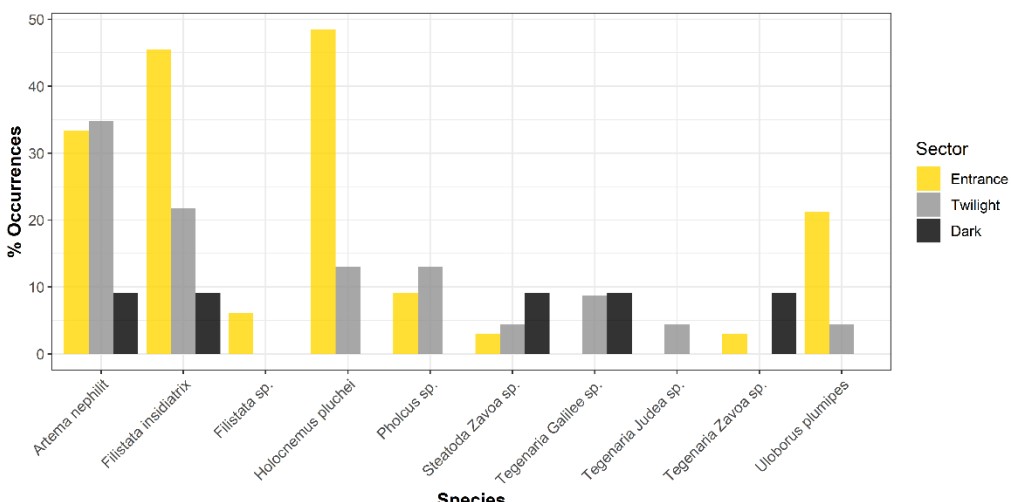

**Figure A1.** Bar plot of the percentage of surveys in which the 10 species with significant associations to cave zones were identified across cave zones. Dark, twilight, and entrance abundances are denoted by black, grey, and yellow, respectively.

**Table A1.** Significant univariate MGLM results for the 10 species with significant associations to cave zones; deviance and probability are given for each. Other significant variables are listed, as well as interactions between zone and those variables. For the other associations, the nature of that association is given as "+" or "−" for positive or negative associations, respectively, and "N", "C", and "S" are given for prevalence in north, central, or southern geographical regions, respectively. The "category" column denotes whether that species is troglobite, troglophile, or accidental (see Gavish-Regev et al., 2021 [19] for the assignment of species to categories). The "dominant zone" column denotes which ecological zones(s) a species was mostly commonly found in. The abundance of these spiders across different cave zones is visualised in Figure A1.

| Species | Category | Dominant Zone | Dev | *p* | Other Associations | Interactions |
|---|---|---|---|---|---|---|
| *Artema nephilit* | Troglophile | Entrance/Twilight | 31.996 | 0.001 | + Temperature (Dev = 31.996, *p* = 0.001) − Elevation (Dev = 105.579, *p* = 0.001) | - |
| *Filistata insidiatrix* | Troglophile | Entrance | 35.957 | 0.001 | − Temperature (Dev = 22.451, *p* = 0.001) − Elevation (Dev = 52.873, *p* = 0.001) N Region (Dev = 69.439, *p* = 0.001) | Zone/Region (Dev = 11.065, *p* = 0.062) |
| *Filistata* sp. | Troglophile | Entrance | 31.16 | 0.001 | C Region (Dev = 43.446, *p* = 0.001) | |
| *Holocnemus pluchei* | Troglophile | Entrance | 49.336 | 0.001 | − Temperature (Dev = 50.909, *p* = 0.001) − Elevation (Dev = 50.702, *p* = 0.001) | Zone/Region (Dev = 12.158, *p* = 0.043) |
| *Pholcus* sp. | Troglophile | Entrance/Twilight | 11.861 | 0.066 | C Region (Dev = 31.48, *p* = 0.001) | - |
| *Steatoda* Zavoa sp. | Troglophile | Dark | 12.275 | 0.058 | + Temperature (Dev = 51.772, *p* = 0.001) + Elevation (Dev = 71.376, *p* = 0.001) | - |
| *Tegenaria* Galilee sp. | Troglobite | Twilight/Dark | 34.444 | 0.001 | − Temperature (Dev = 9.696, *p* = 0.068) + Elevation (Dev = 29.507, *p* = 0.001) N Region (Dev = 31.32, *p* = 0.001) | Zone/Temperature (Dev = 53.994, *p* = 0.001) |
| *Tegenaria* Judea sp. | Troglobite | Twilight | 17.107 | 0.006 | − Temperature (Dev = 17.107, *p* = 0.006) C Region (Dev = 16.456, *p* = 0.005) | - |
| *Uloborus plumipes* | Accidental | Entrance | 22.425 | 0.001 | C Region (Dev = 29.468, *p* = 0.001) | - |
| *Tegenaria* Zavoa sp. | Troglophile | Dark | 39.491 | 0.001 | + Temperature (Dev = 158.41, *p* = 0.001) | - |

*Appendix A.3. Discussion*

The differences in assemblage structure between zones will mostly be affected by those taxa individually associated with different zones (highlighted in Table A1). These species

belong to just a few families (Filistatidae, Pholcidae, Theridiidae, Agelenidae, and Uloboridae). The two most common families among these—Pholcidae and Agelenidae—displayed diametric consistency in their cave zone occupancy. Pholcidae were most prevalent in the entrance zones (being troglophiles), with a consistent presence in the twilight zones, and only one species (*Artema nephilit*) appearing in the dark, albeit rarely. Agelenidae, however, appeared across all three zones, with the three highlighted species (undescribed species found in Galilee, Judea, and Zavoa) found in different permutations of the cave zones, indicating a high degree of generalism for cave zones. Most of the species significantly differing in their zone occupancy were troglophiles, with the only two not found in the cave entrance being the two troglobite species (*Tegenaria* sp. from Galilee caves and *Tegenaria* sp. from a cave in the Judea), and the only accidental species (*Uloborus plumipes*) occurring almost exclusively in the entrance.

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
