# Peer review of "It’s All about the Zone: Spider Assemblages in Different Ecological Zones of Levantine Caves"

_diversity, doi:10.3390/d13110576_

Round 1
Reviewer 1 Report
In the revised article, submitted to MDPI Diversity, Authors have presented results of interesting study on spider assemblages in caves.
Generally, I find the results valuable as they include 1) caves as specific habitats, 2) various ecological zones 3) measurements of abiotic conditions. Generally, the manuscript is interesting, however in my opinion the text requires improving. I hope the manuscript will gain after revision. The introduction presents background of the study and current knowledge, and the hypotheses. Material and methods are clearly described, and majority of the results are fine. I have two issues for consideration.
First, I had problems with data on figures as they don’t have letters. They are of course readable, but it would be easier to find description in figure caption having letters A-C or A-D.
Second it is weird to find alternative results in appendix. I understand that Authors wanted to provide detailed results but in my opinion the Readers may be confused reading the text. I am affaid that readers will not understand results in that case. Minor comments are presented in the attached manuscript file.

Author Response
We would like to thank the three reviewers and the editorial team for an insightful and valuable round of review. We are very pleased that the reviewers recognized the value of our work and below we address each of their comments individually.
Reviewer 1:
In the revised article, submitted to MDPI Diversity, Authors have presented results of interesting study on spider assemblages in caves.
Generally, I find the results valuable as they include 1) caves as specific habitats, 2) various ecological zones 3) measurements of abiotic conditions. Generally, the manuscript is interesting, however in my opinion the text requires improving. I hope the manuscript will gain after revision. The introduction presents background of the study and current knowledge, and the hypotheses. Material and methods are clearly described, and majority of the results are fine. I have two issues for consideration.
First, I had problems with data on figures as they don’t have letters. They are of course readable, but it would be easier to find description in figure caption having letters A-C or A-D.
We have added these to all relevant figures.
Second it is weird to find alternative results in appendix. I understand that Authors wanted to provide detailed results but in my opinion the Readers may be confused reading the text. I am afraid that readers will not understand results in that case.
We appreciate the views of the reviewer. Indeed, the manuscript is more complex as a result of the supplementary inclusion of an alternative analysis; however, we believe that this additional perspective is valuable in representing the whole collected dataset. We understand that this supplementary analysis is affected by potential autocorrelation and pseudoreplication, but it addresses any potential criticism that our cave zone selection criteria in the other model (i.e., taking only the deepest zone of each cave) may have biased our results. We believe that this alternative analysis offers value, even if not appropriate for the main text.
Minor comments are presented in the attached manuscript file.
These have been addressed by changing the text accordingly, with the exception of two comments. The reviewer suggests moving a sentence from Figure 3’s legend into the materials and methods; we have not done this simply because this is a description of the visualised data to aid interpretation, rather than a methodological approach. This reviewer also identified some species names that were not italicised; this was not the case in our submitted document so we expect it is an artefact of the document upload process and can be easily remedied in copy-editing if it persists.

Reviewer 2 Report
The article “It's all about the zone: spider assemblages in different ecological zones of Levantine caves” is a great work that addresses the diversity of spiders in cave environments, an ecosystem often neglected by spider studies. You just need to improve the quality of writing with attention to the English language, and also the images of the figures.Author Response
We would like to thank the three reviewers and the editorial team for an insightful and valuable round of review. We are very pleased that the reviewers recognized the value of our work and below we address each of their comments individually.
Reviewer 2:
The article “It's all about the zone: spider assemblages in different ecological zones of Levantine caves” is a great work that addresses the diversity of spiders in cave environments, an ecosystem often neglected by spider studies. You just need to improve the quality of writing with attention to the English language, and also the images of the figures.
Since the first and fifth authors of this manuscript are, indeed, native English speakers, all authors have a demonstrably excellent grasp of the English language, and no specific examples of poor grammar were given, we reviewed the manuscript again and changed the style in places we found fit.

Reviewer 3 Report
This is an interesting study in which spider species assemblages were surveyed at different depths (zones) in 35 Israeli caves and the results subjected to multivariate statistical analyses. Spider assemblages were predictably found to differed between entrance, twilight and dark zones, with ‘troglophiles’ and accidentals occupying the cave entrance, troglobites occupying the dark zones, and hybrid assemblages in the twilight zones. Entrance zone assemblages were similar among different caves and comprised relatively common species also found in epigean habitats, whereas dark zone assemblages were relatively distinct, consistent with adaptation and speciation and of cave-endemic troglobites in the dark zone.
There are few substantive problems with the manuscript, but the following points should be considered in the revision.
As noted on p. 2, the terms ‘Levant’ and ‘Levantine’ refer to a large area in the Eastern Mediterranean. In its narrowest sense, this includes present-day Syria, Lebanon, Jordan, Israel, Palestine and most of Turkey southwest of the middle Euphrates. However, the study was restricted to Israel/Palestine. Therefore, use of ‘Levant’ and ‘Levantine’ in title and elsewhere seems imprecise and, frankly, misleading. Why not simply refer to ‘Israel/Palestine’ and ‘Israeli/Palestinian’?
Correct use of the Schiner-Racovitza terminology is critically important for a paper on cave biota. However, the definition and usage of the term ‘troglophile’ (defined on p. 2 as ‘organisms with strong affinity to caves’) is inconsistent with recent recommendations. The term ‘trogloxene’, which is not even mentioned in the manuscript, should be used to refer to ‘cave entrance assemblages composed of relatively common species that can be found also in epigean habitats.’ See citation 10 (Trajano & Carvalho, 2017) for the correct application of these terms and revise the manuscript accordingly.
Voucher specimens, documenting identifications, are essential for the repeatability of community ecological studies. Although reference is made to another paper, Gavish-Regev et al., it would be helpful to provide brief details about where the vouchers are deposited and how they were identified.
The statistical analyses used in the study assume each species is an independent data point. However, that may not necessarily be the case, especially when congeners, which may share a common ancestor, are involved. Have the authors considered if or how this may impact the study? It should, at least, be mentioned.
Parts of the Abstract and Discussion overstate the findings concerning adaptation and speciation in caves, for example, on p. 11: ‘This would suggest that diversity radiates from the cave entrance, likely through speciation from relict ancestor species. … low nestedness is likely to be a result of the speciation of relict species and low gene flow between caves and cave zones [1]. These processes likely propagated a relatively rapid transition from entrance assemblage species to distinct dark zone troglobite species, the former in some cases becoming extinct in the epigean system and cave entrances. Dark zone assemblages are thus likely to have been developed from entrance and twilight assemblages but differentiated from them over time through these speciation events. Dark zone spider assemblages may have been formed through the initial adaptive shift of species with pre-adaptations into caves, such as those adapted to shallow trenches or hollows [34], facilitating further adaptation into dark zones, under the adaptive shifts hypothesis (ASH; [35]). Alternatively, these assemblages may have been formed under the climatic relict hypothesis (CRH) which suggests that ancestral and/or relict species, that do not possess pre-adaptations, were pushed into caves due to changes in climatic conditions of the epigean habitat that was no longer suitable for them.’
Let us not lose sight of what is or what is not presented in the manuscript. Data are presented on different assemblages of species at different zones in a range of caves, at a point in time. No historical phylogenetic, evolutionary or biogeographical reconstruction, of any kind, is presented. Therefore, whereas the findings on spider assemblages at different zones in different caves may be *consistent with* various hypotheses about evolution in caves, that’s as far as it goes.
In the same vein, are there data or citations to substantiate the statement that a continuum of ecological zones, differing in their microhabitat conditions can ‘propagate relatively rapid speciation and diverse assemblages of highly specialized spider fauna’ (p. 1)?
Many Latin names are not italicized, e.g., on pp. 7, 8, 10, 15, 17, 18.
Author Response
We would like to thank the three reviewers and the editorial team for an insightful and valuable round of review. We are very pleased that the reviewers recognized the value of our work and below we address each of their comments individually.
Reviewer 3:
This is an interesting study in which spider species assemblages were surveyed at different depths (zones) in 35 Israeli caves and the results subjected to multivariate statistical analyses. Spider assemblages were predictably found to differed between entrance, twilight and dark zones, with ‘troglophiles’ and accidentals occupying the cave entrance, troglobites occupying the dark zones, and hybrid assemblages in the twilight zones. Entrance zone assemblages were similar among different caves and comprised relatively common species also found in epigean habitats, whereas dark zone assemblages were relatively distinct, consistent with adaptation and speciation and of cave-endemic troglobites in the dark zone.
There are few substantive problems with the manuscript, but the following points should be considered in the revision.
As noted on p. 2, the terms ‘Levant’ and ‘Levantine’ refer to a large area in the Eastern Mediterranean. In its narrowest sense, this includes present-day Syria, Lebanon, Jordan, Israel, Palestine and most of Turkey southwest of the middle Euphrates. However, the study was restricted to Israel/Palestine. Therefore, use of ‘Levant’ and ‘Levantine’ in title and elsewhere seems imprecise and, frankly, misleading. Why not simply refer to ‘Israel/Palestine’ and ‘Israeli/Palestinian’?
We wish to take a stance of political neutrality. Given the sensitive nature of the political situation between Israel and Palestine and the distribution of our sampling across a large geographic area in this region, we have associated our surveys with the Southern Levant (itself effectively synonymous with Israel/Palestine) to mitigate any controversy. Whilst our surveys do not represent the entire Levantine region, they do represent a sample within this region, and we thus do not believe that our use of the term is in any way misleading or inaccurate. However, we did consider the comment and changed “Levantine caves” to "35 caves in Israel and Palestine in the initial description of the sampling to increase clarity and changed in some places “Levantine” or “Levant” to “southern Levant”.
Correct use of the Schiner-Racovitza terminology is critically important for a paper on cave biota. However, the definition and usage of the term ‘troglophile’ (defined on p. 2 as ‘organisms with strong affinity to caves’) is inconsistent with recent recommendations. The term ‘trogloxene’, which is not even mentioned in the manuscript, should be used to refer to ‘cave entrance assemblages composed of relatively common species that can be found also in epigean habitats.’ See citation 10 (Trajano & Carvalho, 2017) for the correct application of these terms and revise the manuscript accordingly.
Importantly, these descriptors, as with many terms in ecology, differ in the nuance of their application by different researchers. On page 2 we changed the explanation of troglophiles from “organisms with strong affinity to caves” to: “species that have source populations both in hypogean and epigean habitats”, cited Trajano & Carvalho, 2017 and added an additional citation:
- Trajano, E. Ecological Classification of Subterranean Organisms. In Encyclopedia of Caves; Elsevier, 2012; pp. 275–277 ISBN 978-0-12-383832-2.
Additionally, we added the following statement everywhere relevant that “see Gavish-Regev et al. 2021” was written: “for the assignment of species to categories”.
We changed the term “visitor” to “accidental” for consistent use. However, we decided not to use the term “trogloxene”, as per Trajano, 2012:
1) “It is noteworthy that troglobites, troglophiles and trogloxenes are all subterranean, i.e., they are all adapted to subterranean life, each in their own way.” (Trajano, 2012)
2) “A regular use of subsurface habitats is the first criterion to distinguish subterranean organisms from accidentals, thus isolated observations are insufficient. Repeated observations, supported by data on distribution, ecology and biology of the taxa of interest, are needed for a conclusive classification into the Schiner-Racovitza system.” (Trajano, 2012)
3) “Trogloxene” would function as sink-populations of epigean source populations” and “Distinction between troglophiles and trogloxenes is not trivial because in both cases individuals move between the subterranean environment and the surface. Evidence of subterranean source populations characterizing the first ones includes the presence of all age/size classes throughout the cave, throughout the annual cycle” and “Moreover, several trogloxenes use caves seasonally, being absent during part of the year. Therefore, a definitive distinction between troglophiles and trogloxenes depends on populational studies conducted on an annual basis.” (Trajano, 2012)
In our system “accidental” species did not regularly use caves, and “troglophiles” had both epigean and hypogean populations with all age and size classes found throughout the cave and year.
Voucher specimens, documenting identifications, are essential for the repeatability of community ecological studies. Although reference is made to another paper, Gavish-Regev et al., it would be helpful to provide brief details about where the vouchers are deposited and how they were identified.
We have added the following statement to the end of the survey methods section to address this:
“Voucher specimens were deposited in the National Arachnid Collection in the National Natural History Collections of The Hebrew University of Jerusalem (NNHC, HUJI).”
The statistical analyses used in the study assume each species is an independent data point. However, that may not necessarily be the case, especially when congeners, which may share a common ancestor, are involved. Have the authors considered if or how this may impact the study? It should, at least, be mentioned.
Taxonomy is, indeed, a valuable predictor of habitat occupancy and overall ecology since evolutionary constraints dictate species assemblages. The analysis of closely related species as distinct discrete assignments within a single assemblage is, however, commonplace in community ecology. Primary survey analyses such as this, for which taxonomic information is often poorly established, are valuable nonetheless and, in fact, can help in the description of undescribed species. The as-of-yet undescribed species (notably Tegenaria spp.) in this study are currently being described (each already decidedly a separate species, most being new to science), but the associated publications are not yet finalised and submitted. Their distinct assignments in these descriptions are consistent with those presented in this manuscript.
Parts of the Abstract and Discussion overstate the findings concerning adaptation and speciation in caves, for example, on p. 11: ‘This would suggest that diversity radiates from the cave entrance, likely through speciation from relict ancestor species. … low nestedness is likely to be a result of the speciation of relict species and low gene flow between caves and cave zones [1]. These processes likely propagated a relatively rapid transition from entrance assemblage species to distinct dark zone troglobite species, the former in some cases becoming extinct in the epigean system and cave entrances. Dark zone assemblages are thus likely to have been developed from entrance and twilight assemblages but differentiated from them over time through these speciation events. Dark zone spider assemblages may have been formed through the initial adaptive shift of species with pre-adaptations into caves, such as those adapted to shallow trenches or hollows [34], facilitating further adaptation into dark zones, under the adaptive shifts hypothesis (ASH; [35]). Alternatively, these assemblages may have been formed under the climatic relict hypothesis (CRH) which suggests that ancestral and/or relict species, that do not possess pre-adaptations, were pushed into caves due to changes in climatic conditions of the epigean habitat that was no longer suitable for them.’
Let us not lose sight of what is or what is not presented in the manuscript. Data are presented on different assemblages of species at different zones in a range of caves, at a point in time. No historical phylogenetic, evolutionary or biogeographical reconstruction, of any kind, is presented. Therefore, whereas the findings on spider assemblages at different zones in different caves may be *consistent with* various hypotheses about evolution in caves, that’s as far as it goes.
We have made the overall phraseology of this section much more conditional and, importantly, we have caveated it with a final sentence explaining that we cannot make these assertions without further research in an evolutionary/biogeographical context.
“Cave assemblages show a graded pattern of dissimilarity, with dark and entrance assemblages appearing most dissimilar, and twilight assemblages existing as an intermediate. The relative similarity of entrance assemblages between different caves, compared with the dissimilarity of different dark assemblages (as exemplified in Figure 5), suggests that dark zone assemblage composition is not directly derived from the species pool at the cave entrance, but may have diverged from ancestral entrance assemblages over time. This is supported by a low nestedness of the assemblages within the caves, contrary to our original hypothesis. These patterns of dissimilarity and low nestedness may be due to the existence of relict species in some of the caves sampled, or to speciation within caves as a consequence of low gene flow [1]. Weak gene flow could result in a transition from entrance assemblage species to distinct dark zone troglobite species through adaptive radiation [1,35]. Following the adaptive shift hypothesis (ASH; [38]), troglobitic dark zone spider assemblages could have developed from entrance and twilight assemblages, but differentiated from them over time through these speciation events. These assemblages may have included species with pre-adaptations to caves, such as those adapted to shallow depressions or hollows [37], thus facilitating further adaptation into dark zones. Following the climatic relict hypotheses (CRH), dark zone assemblages may include ancestral or relict species that do not possess pre-adaptations, but survived in caves after changes in climatic conditions rendered the epigean habitat unsuitable for them [36]. The two hypotheses are not mutually exclusive, however, and our study – an ecological snapshot in time – was not designed to test these hypotheses. Further study is required to fully elucidate these evolutionary processes.”
In the same vein, are there data or citations to substantiate the statement that a continuum of ecological zones, differing in their microhabitat conditions can ‘propagate relatively rapid speciation and diverse assemblages of highly specialized spider fauna’ (p. 1)?
We have adjusted the wording to this section to align its meaning more explicitly with the literature, and added two citations:
“Weak gene flow could result in a transition from entrance assemblage species to distinct dark zone troglobite species through adaptive radiation [1,35].”
- Snowman, C. V.; Zigler, K.S.; Hedin, M. Caves as islands: mitochondrial phylogeography of the cave-obligate spider species Nesticus barri (Araneae: Nesticidae). J. Arachnol. 2010, 38, 49–56.
- Wilson, E.O.; O’Brien, R.D.; Susman, M.; Boggs, W.E.; Eisner, T.; Briggs, W.R.; Dickerson, R.E.; Metzenberg, R.L. The Multiplication of Species; Biogeography. In Life on Earth; Sinauer Associates, Inc.: Stamford, USA, 1973; pp. 824–877.
Many Latin names are not italicized, e.g., on pp. 7, 8, 10, 15, 17, 18.
This appears to be a formatting issue when uploading; the files originally submitted did indeed have these species names italicized upon submission, as do those submitted in this round.

Round 2
Reviewer 3 Report
I appreciate the authors' considered responses to the comments and while I disagree with some of their opinions (e.g., 'southern Levant' still fails to accurately characterize the study area, which omits the Sinai Peninsula), I recommend to accept without further revision.